# Holographic Micromirror Array with Diffuse Areas for Accurate Calibration of 3D Light-Field Display

**Lode Jorissen** [1],[†] , **Ryutaro Oi** [2],[†], **Koki Wakunami** [2], **Yasuyuki Ichihashi** [2], **Gauthier Lafruit** [3], **Kenji Yamamoto** [2], **Philippe Bekaert** [1] **and and Boaz Jessie Jackin** [4],[*],[†]

1   Expertise Centre for Digital Media, Hasselt University–tUL–Flanders Make, 3590 Diepenbeek, Belgium; lode.jorissen@uhasselt.be (L.J.); philippe.bekaert@uhasselt.be (P.B.)
2   National Institute of Information and Communications Technology, Tokyo 184-8795, Japan; oi.ryutaro@nict.go.jp (R.O.); k.wakunami@nict.go.jp (K.W.); y-ichihashi@nict.go.jp (Y.I.); k.yamamoto@nict.go.jp (K.Y.)
3   Laboratory of Image Synthesis and Analysis (LISA), Université Libre de Bruxelles/Brussels University, Av. F.D. Roosevelt 50 CP165/57, 1050 Brussels, Belgium; gauthier.lafruit@ulb.ac.be
4   Center for Design Centric Engineering, Kyoto Institute of Technology, Kyoto 606-8585, Japan
*   Correspondence: jackin@kit.ac.jp; Tel.: +81-75-724-7675
†   These authors contributed equally to this work.

**Abstract:** Light field 3D displays require a precise alignment between the display source and the micromirror-array screen for error free 3D visualization. Hence, calibrating the system using an external camera becomes necessary, before displaying any 3D contents. The inter-dependency of the intrinsic and extrinsic parameters of display-source, calibration-camera, and micromirror-array screen, makes the calibration process very complex and error-prone. Thus, several assumptions are made with regard to the display setup, in order to simplify the calibration. A fully automatic calibration method based on several such assumptions was reported by us earlier. Here, in this paper, we report a method that uses no such assumptions, but yields a better calibration. The proposed method adapts an optical solution where the micromirror-array screen is fabricated as a computer generated hologram with a tiny diffuser engraved at one corner of each elemental micromirror in the array. The calibration algorithm uses these diffusing areas as markers to determine the relation between the pixels of display source and the mirrors in the micromirror-array screen. Calibration results show that virtually reconstructed 3D scenes align well with the real world contents, and are free from any distortion. This method also eliminates the position dependency of display source, calibration-camera, and mirror-array screen during calibration, which enables easy setup of the display system.

**Keywords:** computer generated holography; holographic optical element; light field display; 3D display; calibration

## 1. Background

Light field display is one of the promising candidates that is expected to succeed as the next generation 3D display technology [1,2]. They can be used both as a head-mounted device [3,4] or as a head-up display [5,6]. Light field displays usually consist of a 2D display device (which displays integral images) combined with a microlens-array screen. The microlens-array is sandwiched on top of the 2D display device (in case of LCD panel) [7] or is kept at a distance from the display device (in-case of a projector) [8]. A 3D scene is decomposed into a set of 2D integral images which is displayed on the 2D display device. The light rays from this 2D integral-image then passes through a micro lens array screen before reaching the viewers eye. It is the microlens array that recomposes the 2D

integral-images back into a 3D scene, to be visualized. For accurate 3D reconstruction, the light rays from each pixel of the 2D display devices must pass through the exact predetermined point in the microlens-array sheet. In other words, the integral images displayed by the display device must be well aligned with the corresponding elemental lens in the microlens-array sheet [9]. Such an alignment is relatively easy to achieve on a LCD panel based system since the microlens-array is sandwiched on top of the panel, where, in a projector based system, it poses severe challenges since it is kept at a distance from the projector [10]. Considering augmented reality applications, a projection based system is more advantageous compared to an LCD based system [11], and this paper focuses on a projection based system. Given the size of the elemental microlenses in the array, which is less than a millimeter, any small misalignment will cause a visible distortion in the reconstruction. Hence, a dedicated calibration is required to get such a system working, especially when the display source is a projector where the mirror-screen is at a distance.

We have reported the successful design, build, and calibration of such a projection based integral imaging light field system [11]. In this display system, the microlens-array sheet was replaced with a *concave micromirror* array sheet (hereafter to be mentioned as micromirror), to make the system work in reflection mode as shown in Figure 1a. In other words, each concave micromirror replicates the function of the microlens, but in reflection mode where the real focus point falls in front of the screen (on the same side of viewer). The viewer is located on the same side of the projector, as seen from Figure 1a. The main advantage of this configuration comes from the fact that these micromirror arrays can be made to be see-through, by fabricating them as volume reflection holograms, known as Digitally Designed Holographic Optical Elements (DDHOE) [12]. This enables the user to see both the virtual and real contents combined together as shown in Figure 1a. A fully automated technique to calibrate the system mentioned above was reported by us [13]. The method worked well without any visible distortions but demands precise positioning of camera, display, and DDHOE screen due to the assumptions used in the calibration procedure. In addition to that, the earlier method was not capable of aligning the reconstructed virtual 3D contents with the real world objects. In other words, the projected virtual contents won't overlap exactly on top the real world contents, from all viewing angles. Other calibration methods reported were also affected by similar issues [14]. In this report, we propose a solution by resorting to an optical approach which is to add tiny diffusers to one corner of each micro concave-mirror in the DDHOE screen. The new calibration method is then anchored on detecting these diffuse areas for accurate calibration. In other words, these diffuse areas serve as markers to determine the absolute position of each elemental micro concave-mirror irrespective of the position of projector and DDHOE screen. The computer generated holographic technique was used to fabricate both the concave-mirrors and diffusers on a single hologram sheet.

In this paper, first, we will discuss why the assumptions are not a good idea if an accurate alignment between the virtual scenes and the real world is required. Next, we introduce a solution that adds small diffusive markers to the screen that will serve as a reference. Finally, we compare the light field reconstruction that uses the calibration approach introduced in [13] with a reconstruction that utilizes the newly introduced diffusive markers for calibration.

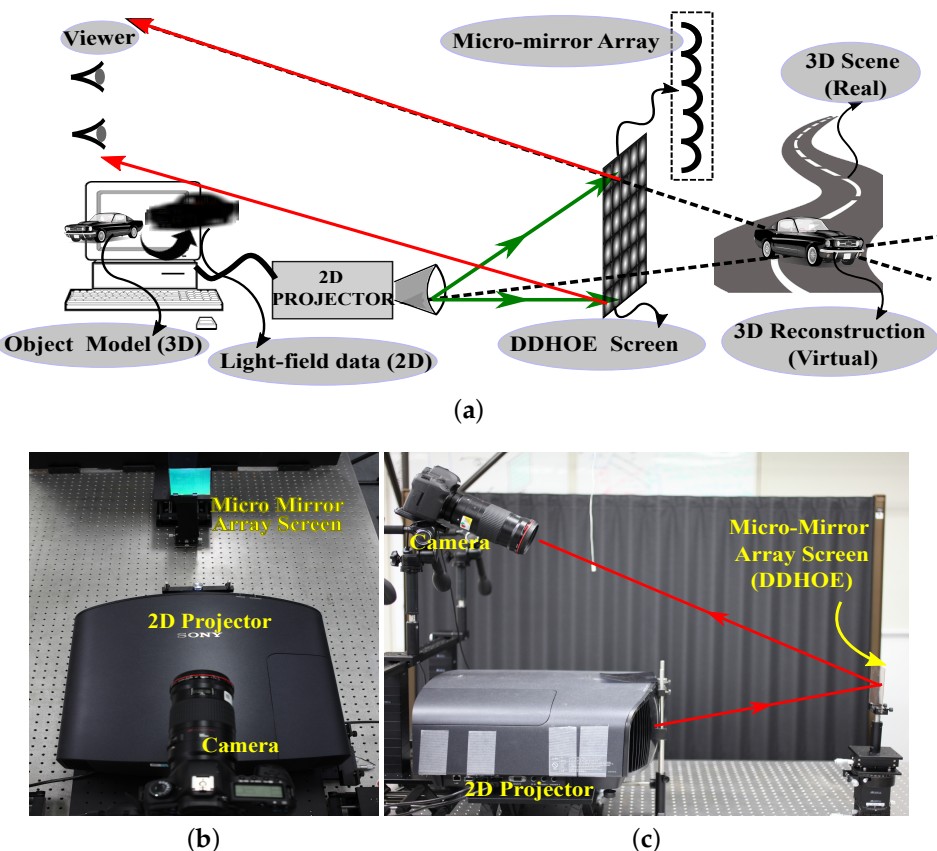

**Figure 1.** (**a**) schematic of the light-field display system; (**b**) photograph of the top-view of the real display system; (**c**) photograph of the side view of the real display system.

## 2. Calibration Inaccuracies

In order to calibrate a projection based light field display system, the DDHOE screen is placed at the projector's focus and is lit by the projector as shown in Figure 2a. A camera is placed at the position where the user will be located during playback. The camera captures light bouncing back from the DDHOE screen (Figure 2a). By projecting a structured light pattern (graycodes in this case), the relation between projector pixels and micromirror position can be determined. The main assumption used in the calibration reported previously is that the brightest spots of the micromirrors were at the centers of the micromirrors (shown as rays of peak intensity in Figure 2a). In reality, this assumption does not hold and can result in a shift of the projected scene with relation to the real-world objects.

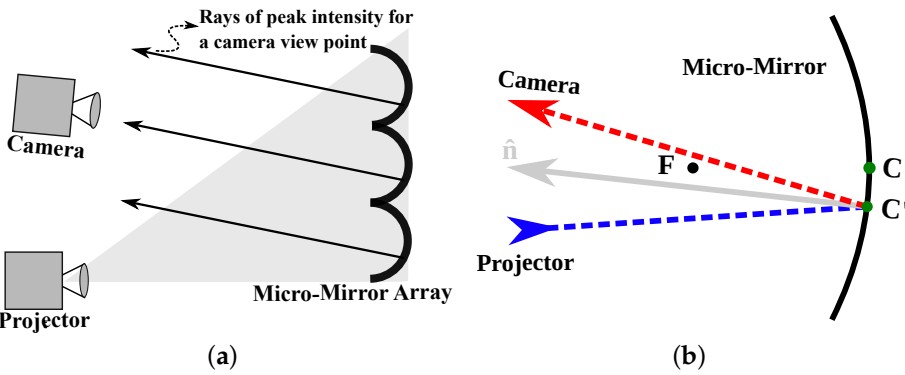

**Figure 2.** (**a**) schematic of the calibration configuration; (**b**) difference between the observed point $C'$ and the actual center $C$ of the mirror. The position of the observed point is dependent on the focal length of the mirror and the direction of the projected ray, as well as the surface normal $\hat{n}$.

In fact, as can be seen in Figure 2b, the observed point $C'$ corresponds to the location at which the light ray that hits the camera sensor, is reflected. This means that there is an offset between the observed point $C'$ and the actual center $C$ of the mirror. It is to be noted that the curved surface represented in Figure 2b corresponds to a virtual curvature, whereas the real hologram surface is flat. In other words, the light rays that are reflected off the hologram surface behaves as if they are getting reflected off of a curved surface. Thus, the surface normal $\hat{n}$ here is the normal to the virtual curvature at the point of incidence. In addition to that, the optical axis of each optical mirrors are tilted by a small angle [11]. Hence, the offset is dependent on several parameters such as the focal length and tilt of the corresponding micromirror, as well as the positioning of both the projector and the camera used for calibration. By ignoring this shift and assuming that the centers of the projected elemental images should be mapped onto the observed brightest point for the corresponding micromirror, two issues are introduced:

1.  View Shift: since the observed points can only be noticed at the position of the calibration camera, the calibration procedure will map the central view of the scene to the position of the camera. This means that, if the camera is located to the left of the viewing zone's center the views of the scene will all be shifted to the left as well. As a result, some views cannot be perceived at the left side of the viewing zone but will be visible on the right side instead.
2.  Distortion: although, the screen handles the divergence of the projector [11] (if positioned correctly), the camera will still observe a converging set of rays being reflected from the screen. Because of this, the observed points will not be located at the same position within each micromirror. This introduces a distortion since the calibration will try to minimize the reprojection error as much as possible, resulting in an overcompensation of the lens distortion parameters.

Assuming that the projector is located precisely on a designated position, one can mitigate these issues by precise camera placement. To remove the shift of the view, one can place the camera at the center of the viewing zone as precisely as possible. This makes the system reflect the center view towards the center of the zone. In order to reduce distortion, one can place the camera at a larger distance from the screen. This will make sure that the light rays captured by the camera become more parallel to each other, and thus the position at which they intersect on the micromirrors becomes more similar. Even though these adjustments reduce the impact, they introduce several difficulties in handling the display and also do not completely remove the issue. These shifts and distortions, after calibration, may also be observed when taking a closer look at the DDHOE while projecting a 2-color integral image (presented in Section 4).

As we could see in the experiments from the previous paper, none of these issues are directly visible when one just wants to show a 3D object. However, they can become an issue for augmented reality applications, where an accurate alignment between the projected scene (virtual content) and the real-world environment is required.

It is also possible to take a mathematical approach to solve the issue by calculating the amount of shift. By measuring the direction of the incoming and outgoing light rays, one may determine the shift based on the normal around which the light ray reflects. However, this would require a highly accurate calibration of the internal parameters of both camera and projector, as well as the relative positioning of the camera w.r.t. the projector. A small error in these parameters can cause a large error in the calculated shift, especially when working on this scale. Different materials for the DDHOE may also have different properties, and could impact this calculation even further. A short overview of such an approach is given in Appendix A.

In conclusion, to overcome all the difficulties mentioned above, we resorted to an optical solution, which would reveal the absolute position of each elemental micromirrors irrespective of the positioning of projector, calibration-camera, and DDHOE. The idea is to introduce small diffuse markers on the DDHOE screen that will serve as a ground-truth anchor to determine the on-screen location of the brightest points.

### 3. Diffusive Markers Based Compensation

The micromirrors are fabricated as computer generated holograms (CGH) on a photopolymer sheet. While calculating the CGH, small diffuse areas were added to the top-left corner of each micromirror. The diffuse area consists of random dots of size 0.6 μm occupying an area of 150 × 150 μm. The dot size of 0.6 μm is the result of each pixel of the SLM (size 6 μm) being de-magnified 10× on the photopolymer. The dimensions of the diffuse area were experimentally determined so that they could be detected by the dedicated calibration procedure but remain invisible during the reconstruction. Each elemental micromirror in the array is rectangular in shape and is 1 mm × 0.5 mm in dimension. The size of each elemental micromirror was also determined experimentally, so that the best diffraction, resolution, and view-angle was achieved, given the limitation of the fabrication setup. The ratio of length to width of each micromirror was determined in accordance with the aspect ratio of the SLM. The array contains 100 × 200 elemental micromirrors and the whole DDHOE screen is of size 10 cm × 10 cm. The optical axis of each elemental micromirrors are tilted a bit towards the center, in-order to obtain maximum field-of-view [11]. The hologram pattern was generated on the computer, phase wrapping a summation of two terms, (i) a quadratic phase term (that corresponds to the focal length of the lens) and (ii) a linear phase term (that corresponds to the tilt to optical axis) which results in a Freznel-zone like pattern as shown in Figure 3. A detailed explanation to the hologram computation procedure can be found in [11]. These computed digital data are then converted into light modulations by displaying them on a spatial light modulator (SLM) and illuminating it with a laser beam. The light pattern generated by the SLM is then optically de-magnified and transferred to a photopolymer film. The photopolymer used was Bayfol HX-102 manufactured by Covestro. This fabrication process is called as "wave-front printing" and is explained in detail by Wakunami et al. [12]. Figure 4a shows the photograph of a fabricated micro mirrorarray sheet when illuminated with white light from a projector. Since a green laser of wavelength 532 nm was used for fabricating the volume hologram, and only green light reflects back even illuminated with white light (due to the property of wavelength selectivity of a Bragg hologram). The fabricated Bragg holograms had a diffraction efficiency of 60%. It is worth noting here that a wave-optics based treatment is used to define and execute the hologram computation and fabrication process [11], whereas the calibration process to follow will be treated using geometric optics principles.

Due to the light diffusing property, the marker itself is hard to see with the naked eye, as can be observed from Figure 4a. Even when it is fully lit by the projector, it does not interfere with the reconstructed light field: it only reflects a small amount of light towards the observer, compared to the rest of the mirror. This clearly shows that the presence of diffuse areas does not create any effect during reconstruction. A small drawback of this approach is that the viewpoints in one of the corners of the viewing zone will not be reconstructed; however, an observer usually does not reside in these corners of the viewing zone due to practical reasons: the regions are located too high or low for a pleasant viewing experience, as well as the scene that starts to disappear at the border of the viewing zone. The diffuse areas are visible as tiny spots on the bottom-right corner of each mirror-array.

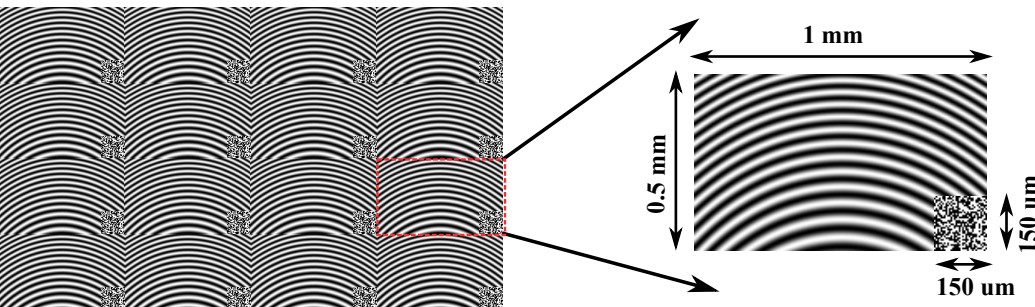

**Figure 3.** Illustration of the interference pattern used by the wavefront printer. The randomness in the lower-right corner introduces a diffusing area in the final micromirror.

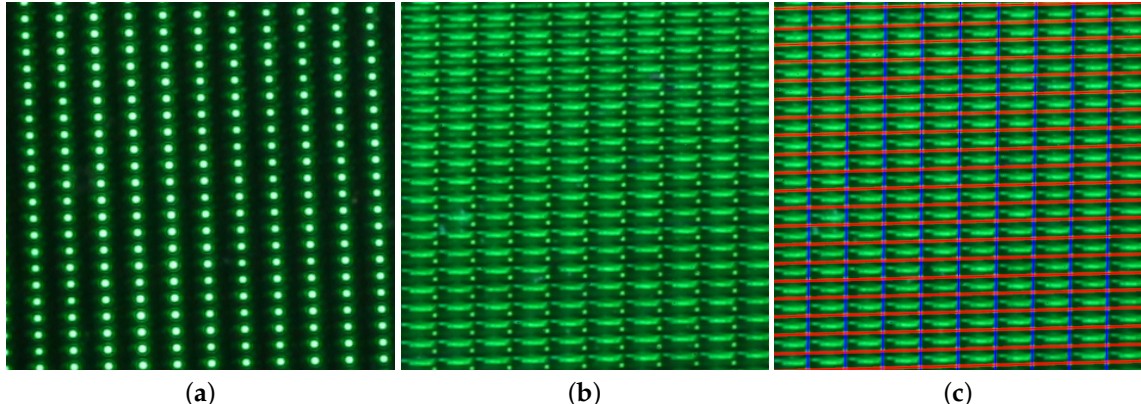

**Figure 4.** (**a**) camera image of the mirror array screen as seen by the naked eye when completely lit by the projector (diffuse areas are not visible); (**b**) camera image of the mirror array screen, captured using a dedicated procedure to detect diffusing markers (diffuse areas are visible as dots); (**c**) borders of the elemental images overlaid on top of the image using the estimated homography. Intersections coincide with the diffusers, which are located at the corners of the diffusers.

In order to use the markers for the calibration, we need to detect them first. A normal, single-shot, picture (shown in Figure 4a) is not sufficient to recognize the markers. Instead, we use a dedicated procedure where the camera is configured to have a long shutter-time (30 s) and a low light sensitivity (ISO 100). Furthermore, a small aperture is used to reduce the incoming light further. While the camera is taking the picture (shutter open), we move a light source around in front of the screen. This way, we integrate the light hitting the diffusers from a different direction, over time. The light reflected from the mirrors is spread over multiple locations on the mirror, which ensures that these reflections do not result in an overly bright image. A crop-out of such an image is shown in Figure 4b. In this image, the markers are clearly visible as small dots. To detect these dots, we first apply an adaptive thresholding algorithm to determine the pixels that received a notable amount of light reflected by the screen. The thresholding algorithm is based on the work of [15] and applies binary thresholding on the image, using a dynamic threshold that is determined on the neighborhood of the pixel to threshold. After thresholding, we apply connected components labeling to determine pixels that belong to the same blob. The connected components are filtered by the area and aspect ratios of the components: the components that represent a diffuser are usually a few pixels wide and high, and have an aspect ratio close to 1. This filter removes the light paths formed by the moving light source being reflected by the mirrors, as the corresponding components tend to be wider than they are high. The remaining components are considered to be the artificially introduced markers.

After detecting the markers, we apply the homography based identification that we described in [13] on the positions of the markers to identify to which mirror a marker belongs. This method utilizes the grid-structure combined with a homography to identify the markers, and filters out possible outliers. Since we know the actual positions of the markers on the screen, contrary to an assumed center, we can use this identification to build a homography $H$, which is a $3 \times 3$ matrix that transforms pixels in the camera image plane to their corresponding location on the DDHOE plane. When visualizing the homography, by drawing the lines corresponding to the elemental images—their borders—the intersections of these lines will coincide with the diffusive markers, as can be seen in Figure 4c.

Finally, we apply our structured light approach from [13] to find correspondences between the projector $(p_x, p_y)$ and camera $(c_x, c_y)$. Several binary patterns are projected onto the DDHOE, allowing us to identify which projector pixels are reflected to which camera pixel.

The homography $H$ is used to transform the observed camera pixels $(c_x, c_y)$, back to the DDHOE plane. Doing so, we obtain the correct position $(X, Y)$ on the DDHOE screen, where the projector pixel is being projected to:

$$\begin{bmatrix} M_x \\ M_y \\ M_z \end{bmatrix} = H \cdot \begin{bmatrix} c_x \\ c_y \\ 1 \end{bmatrix} \tag{1}$$

$$(s_x, s_y) = (M_x/M_z, M_y/M_z); \tag{2}$$

this is the part where we get rid of the assumption that the observed correspondences are located at the centers of the micromirrors, since the previous approach selected $s$ to be the center of the corresponding micromirror. Instead, we now use the calculated location $(s_x, s_y)$ of the correspondence on the DDHOE plane. For each $(s_x, s_y)$, the corresponding projector pixel $(p_x, p_y)$, obtained using structured light, is used for calibration. To calibrate the projector, we utilize a calibration algorithm for cameras, since a projector can, mathematically, be seen as an inverted camera. The calibration optimizes the parameters of the projector, such that for the error between $(p_x, p_y)$ and the projection of $(s_x, s_y, 0)$ onto the projector, its display plane becomes minimal. Parameters to optimize include translation, rotation, focal length, principal point, and distortion. In our implementation, we used the calibration from OpenCV [16]. Finally, we use the obtained parameters to deform the integral image in such a way that the projector optics will undo the deformation. This mapping is achieved by virtually projecting the DDHOE plane, with a correctly aligned integral image, onto the projector with its display plane using the obtained parameters. The resulting image can be displayed by the real projector to obtain the correct light field reconstruction.

## 4. Results

We utilize two approaches to check the accuracy of the proposed calibration method. First, we use a simulation to see how well the projected image maps on the DDHOE screen. We will compare this to a ground-truth reference and to our old calibration from [13] that does not utilize the markers. Finally, we will present light field reconstructions utilizing the real-world display to compare both calibration approaches. To do this, we will align a virtual scene with a 3D-printed object (real scene) that is located at a predetermined position behind the screen. Opposite to a compensation that takes ray directions into account, this approach only requires a rough estimation of the camera's intrinsics, if there is a noticeable lens distortion.

### 4.1. Close-Ups in Simulation

One way to observe the accuracy of the calibration is by projecting a simple two-colored integral image and taking a closer look at the light reflected on a micromirror scale. Figure 5 gives closeups for cases using a ground-truth alignment, no calibration, calibration using the approach from [13], which does not utilize markers, and the approach presented in this section that introduces the transparent diffusive markers. All close-ups are obtained using a simulation program used to develop the calibration software. In our setup, both projector and camera were located at a less than optimal position. The projector was given an extra tilt around its projection axis, while the camera was located 2.5 cm to the left of the viewing zone its center.

In this simulation, we projected a test pattern onto the micromirror array. The used pattern is an integral image in which each elemental image is given a single solid color. These differently colored rectangles help to visualize the correctness of the alignment. In our tests, each elemental image occupied $22 \times 11$ pixels, since the micromirrors are only half as high as they are wide. Most elemental images are colored red or blue to visualize their borders. Some are given a different color (yellow or green) to check whether the mapping between elemental images and micromirrors is correct. Please note that this pattern is not to be confused with a Bayer mosaic. Ideally, the resulting grid should align perfectly with the borders of the rectangular micromirrors, as can be seen in the ground-truth

case. The ellipse-shaped gradients show the regions of the mirrors that reflect light towards the camera. For illustration purposes, the mirrors were made more diffuse than the actual real-life material. The intensity corresponds to the amount of light being reflected in the direction of the camera for that point. A sudden cut-off of the circle corresponds to a transition to a neighboring mirror: e.g., the light at the left side of a micromirror is reflected to a completely different direction than the light at the right side. The same goes for bottom and top. The elemental images in a different color (e.g., green and yellow) helps to determine whether the elemental images project on the correct micromirror.

In the case of the ground-truth, we can see that each micromirror has a single color for the whole micromirror. This indicates that the setup is accurately calibrated and has no distortion. In the case of the uncalibrated setup, most mirrors show a combination of more than one color, indicating that two or more elemental images are projected on the same micromirror. In some cases, micromirrors turn blue where in the ground-truth case they are red. This indicates that the elemental image is being projected on the wrong micromirror. This is reinforced by the fact that neither the green nor the yellow elemental images are projected on their corresponding micromirrors, and are even not visible at all in the provided close-ups. The borders of the test pattern its elemental images are also not parallel to the borders of the micromirrors, since the projector is also slightly tilted around its projection axis.

In the case that we use the calibration without diffusers, the grid lines are parallel to the borders of the mirrors. This shows us that the projector tilt has been compensated for. However, there is a clear shift of the grid. This shift is caused by the fact that the camera was not located at the viewing zone's center, but towards the left of it. This shift will cause the observer to see the opposite side of the scene when moving to the border of the viewing zone. The highlighted yellow and green elemental images roughly project on the same micromirror as in the ground-truth case, proving that the micromirror and elemental image mapping is correct.

Finally, when looking at the calibration that utilizes the diffusers, we can see that the observed image closely resembles the ground-truth: the grid lines of the integral image are parallel to the borders of the micromirrors and each micromirror reflects a single color. That color also corresponds to the one indicated by the ground truth, giving an indication that the elemental images map correctly onto the corresponding mirrors. As with the previous case, the highlighted elemental images map on the correct micromirror.

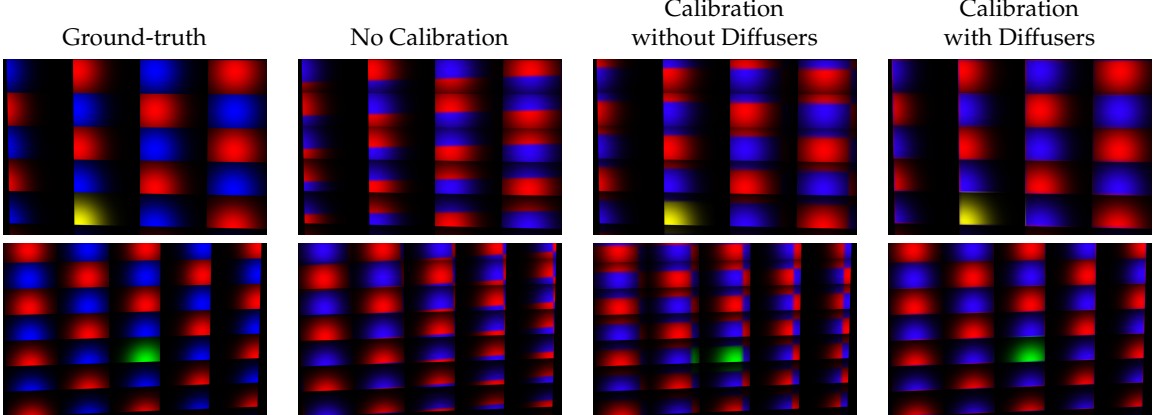

**Figure 5.** Close-ups of a simulated DDHOE while projecting the test pattern. The first column represents the ground-truth. The second column represents the case in which no calibration is applied. The third column represents the calibration method which does not use the diffuse markers. The last column is the result of the proposed calibration which does utilize the diffuse corners. The yellow and green cells indicate that the identification of the micromirrors is correct in both calibration methods. The results from the proposed approach match well with the ground-truth.

### 4.2. Light Field Reconstruction

To test the performances of the diffusive marker approach on a real system, we will look at the actual reconstructed image from the display system shown in Figure 1a–c. To understand the divergence suffered by the reconstructed beam on interaction with the hologram screen, one may refer to Jackin et al. [11]. The reconstructed images were captured using a camera placed in the view zone as shown in Figure 2a. Figure 6b–e shows reconstructions for calibrations using both approaches.

In this experiment, we moved the camera, prior to calibration, about 10 cm to the left of the center of the screen's viewing zone, to imitate an inaccurate camera placement. The displayed scene (Figure 6a) consists of a ring with four arrows pointing towards the center which is to be reconstructed at 3 cm behind the DDHOE screen. A sphere (ball) is floating about 6 cm in front of the center of the ring which is to be reconstructed at 3 m in-front of the DDHOE screen. When this scene is observed from the front, the arrows should be pointing towards the ball. When the observer moves to the left, the ball should appear in front of the right arrow. Figure 6b,c show views recorded at respectively the left side and the center of the viewing zone for the described scene while utilizing the calibration that does not use the diffusive markers. The location of the camera in the left view (Figure 6b) was the same location as the one during calibration. When we observe the scene from the left side, we can see that the ball is located in the center of the ring. This indicates that we are looking at the central view of the displayed scene, from the left side, which is not correct. When we observed the setup from the center of the viewing zone, the ball touches the left arrow, which indicates that we are looking at a right view of the projected scene. These results reveal the effects of camera positioning offsets during calibration. From this, we can understand that simply assuming the camera to be exactly at the center (where the actual position may be slightly offset) during calibration is not sufficient enough. It also reveals that that during calibration, the central view of the scene, is always camera position dependent, if no additional markers are being used. It is to be noted that the ball appears to be cropped at one side, which is due to the shadow of the 3D content (circle with arrows), casted on the ball.

Figure 6d,e show reconstructions using the proposed calibration that utilizes the diffusive corners. The locations of the cameras are the same as those used to capture images Figure 6b,c, and the position of the camera during calibration was also the same as the one in the previous experiment. When observed from the left (Figure 6d), the ball now overlaps the right arrow. This indicates that we are looking at a left view of the projected scene, while observing the display from the left. When observing the scene from the center of the viewing zone, the ball appears to be in the center of the ring. As such, we are observing the center of the scene. This test indicates that a calibration which uses additional markers makes sure that the reconstructed views are projected in the correct directions, contrary to using a calibration without markers.

In another experiment, we tested the alignment of the reconstructed scene with an actual object (real scene). To do this, we placed a 3D-printed object at about 3 cm behind the screen. The complete setup can be seen in Figure 7. The 3D-printed object consists of a flat ring and an arrow that points to the center of the ring. Additionally, we created a 3D scene containing a ring 3 cm behind the display (Figure 8a). This ring has the same dimensions as the 3D-printed ring. We also positioned a disc 6 cm in front of the center of the ring, which means that the disc is located 3 cm in front of the display. When correctly calibrated, both the virtual ring and the 3D-printed ring align from all viewpoints. Again, we applied both calibration approaches. Figure 8b–d show the reconstruction using the calibration without diffusing markers. One can clearly see that the ring of the virtual scene is not aligned with the ring of the 3D-object. Furthermore, ghosting appears at the far edge of the viewing zone, as information from both left and right views of the projected integral image are being observed. When using the calibration with diffusers, Figure 8e–g, both rings align correctly. The virtual disc that floats in front of the display, undergoes a correct parallax with a relationship to the 3D-printed arrow: in the left view, it appears to the right of the arrow, in the right view, it appears to the left and in the center view the arrow is pointing to the disc. This shows us that the views are also displayed in the

correct directions. These results show that the proposed calibration approach is accurate enough to result in a precise alignment between both virtual and real worlds for Augmented Reality applications.

Although the introduced diffusive markers reduce the viewing areas in the corners, they did not negatively impact the overall quality of the reconstruction, nor did they impact the see-through property (transparency) of the screen.

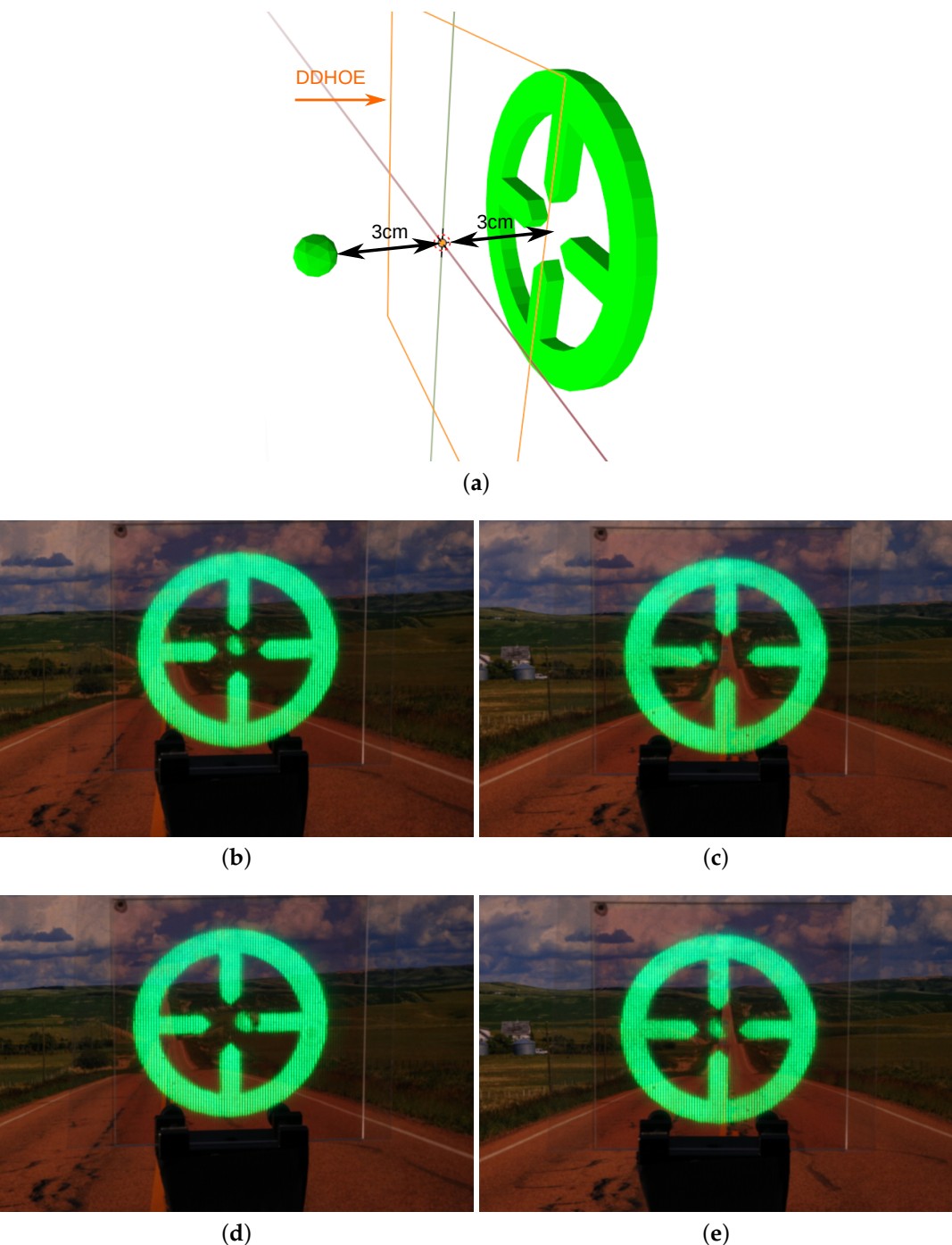

**Figure 6.** (**a**) the scene used in this experiment. Calibration without markers: (**b**) left view, (**c**) center view. Calibration with markers: (**d**) left view, (**e**) center view.

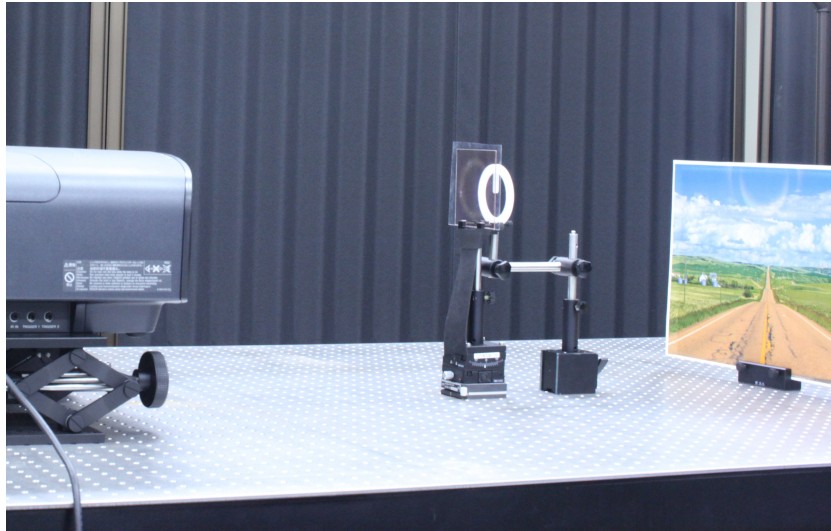

**Figure 7.** To test the alignment of the projected scene with the real-world, we placed a 3D printed object, 3 cm behind the DDHOE display.

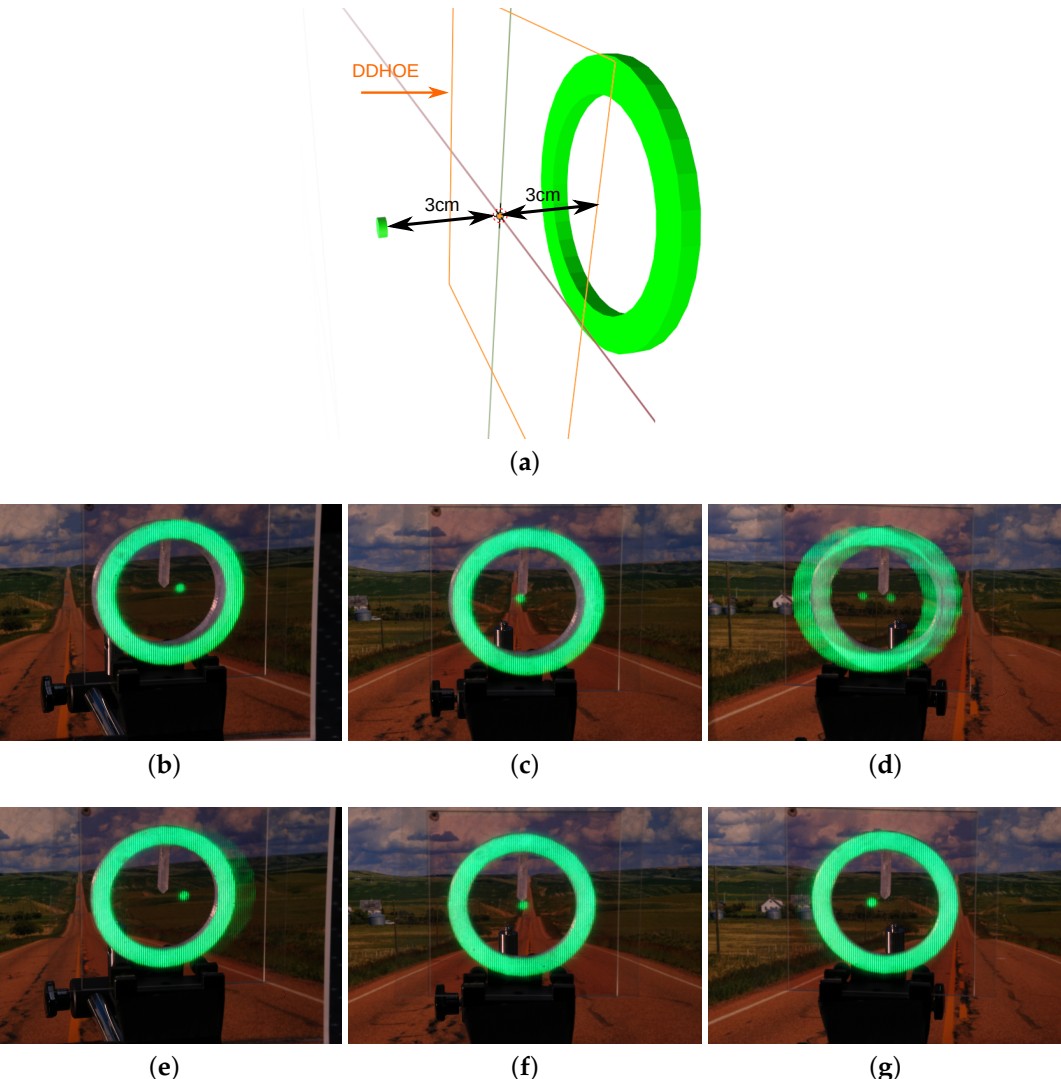

**Figure 8.** Alignment with of a virtual scene (**a**) with 3D-printed object: both circles should align. Calibration without markers: (**b**) left view, (**c**) center view and (**d**) right view. Calibration with markers: (**e**) left view, (**f**) center view and (**g**) right view. The 3D object is located 3 cm behind the DDHOE display.

## 5. Conclusions

In this paper, we introduced a new calibration approach that utilizes diffuse markers, invisible to the naked eye, on the DDHOE screen. Compared to our previous marker-less calibration, this calibration makes sure that the views are sent into the correct direction, without any distortion. It also lets the user freely place the projector, calibration-camera. and DDHOE-screen in the display setup, without having to worry about any positioning errors. Since the approach can be applied on a transparent display, and since it correctly aligns the virtual world with the real world, it proves to be a good calibration approach for when the transparent light field displays are to be used for Augmented Reality applications.

As for now, we foresee a marker in one corner of every micromirror. In future work, we would like to reduce the number of markers, or spread the marker over the four corners of a micromirror. This way, we can further reduce the loss of the viewing zone, or limit the loss to regions from which the observer will likely not look towards the screen. Experiments to measure the aberrations exhibited by the micromirrors and the absolute errors of calibration will also be conducted in the future.

**Author Contributions:** Conceptualization, R.O. and B.J.J.; methodology, L.J.; software, L.J.; validation, L.J. and B.J.J.; formal analysis, L.J.; investigation, L.J. and R.O.; resources, Y.I. and K.W.; writing—original draft preparation, L.J.; writing—review and editing, B.J.J.; supervision, K.Y., G.L. and P.B.; project administration, K.Y. and B.J.J.; funding acquisition, K.Y. and B.J.J. All authors have read and agreed to the published version of the manuscript.

**Funding:** This research was funded by Japan Society For Promotion Of Science (JSPS) Grant No. 18H03281.

**Conflicts of Interest:** The authors declare no conflict of interest.

## Abbreviations

The following abbreviations are used in this manuscript:

DDHOE     Digitally Designed Holographic Optical Element
SLM        Spatial Light Modulator
CGH       Computer Generated Hologram

## Appendix A. Mathematical Compensation

This section gives a short overview of a purely mathematical approach which uses no diffusers. This method works well in the simulation if one has a precise calibration of the camera and projector optics (e.g., focal length, distortion parameters, etc.). In reality, one often cannot achieve these parameters with the required precision.

The mathematical approach assumes that the focal length $f$ of the mirrors are known as well as the directions ($\vec{i}$ and $\vec{o}$) of both the incoming and reflected light rays. These directions can be obtained if we assume that both the camera and projector are intrinsically calibrated and that their relation to each other is known. These calibrations can be obtained by using a calibration toolkit such as the one from Moreno and Taubin [17]. Please note that, by knowing these parameters, we still don't have a calibrated display, as these parameters only describe the relation between a projector and camera and not between the projector and the display. After finding the correspondences using the structured light approach from Section 3, one can calculate both $\vec{i}$ and $\vec{o}$ by deprojecting their corresponding projector and camera pixels.

While the DDHOE represents an array of concave mirrors, the actual surface of the DDHOE is flat. As such, we can model the DDHOE as a flat surface with a surface normal that behaves like the surface normal of a concave mirror. This is illustrated in Figure A1. This property allows us to easily define the calculation of the offset $d$.

Given the incoming ray vector $\vec{i}$ and the outgoing ray vector $\vec{o}$, we can calculate the normal $\vec{n}$ around which the light ray is reflected as follows:

$$\vec{n} = \frac{\vec{o} - \vec{i}}{2} \qquad (A1)$$

the offset $\vec{d}$ can then be calculated by:

$$\vec{d} = \frac{2f\hat{n}_{yz}}{\hat{n}_x} \qquad (A2)$$

where $\hat{n}$ is the normalized vector of $\vec{n}$. $\hat{n}_{yz}$ is the length of $\hat{n}$ after projecting it onto the display plane, while $\hat{n}_x$ is the length of $\hat{n}$ after projecting it onto the display normal. The plane and its normal can be calculated by triangulating the projector and camera correspondences and fitting a plane through the corresponding point cloud.

The actual position of $C'$ on the display can now be calculated as follows: $C' = C - \vec{d}$. In the calibration process from Section 3, we can now use this position $C'$ as a value for $(X, Y)$.

The main drawback of this approach is the requirement of an accurate initial calibration of the intrinsic parameters and the lens distortion of both projector and camera. A small variation in one of these parameters will result in a change of the direction of $\vec{i}$ or $\vec{o}$, causing an incorrect estimate of the offsets. As a result, the light field reconstruction is often still incorrect with a relation to the physical objects. Instead of looking at a mathematical compensation, we looked at a more accurate physical compensation.

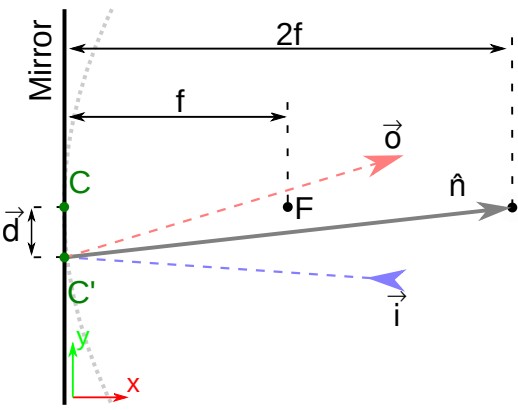

**Figure A1.** Calculation of offset $d$, between the observed point $C'$ and the actual center $C$ of the mirror, given the incoming and reflected rays $\vec{i}$ and $\vec{o}$ and the focal length $f$ of the concave mirror. While the DDHOE represents a concave mirror, the actual surface of the DDHOE is flat. As such, we model the DDHOE as a flat surface with a surface normal $\hat{n}$ that behaves like a concave mirror.

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
