# Peer review of "Holographic Micromirror Array with Diffuse Areas for Accurate Calibration of 3D Light-Field Display"

_applsci, doi:10.3390/app10207188_

Round 1

Reviewer 1 Report

Attached.

Reviewer 2 Report

The authors propose a new method of 3D display alignment, based on the introduction of a special small diffuser on each of the DDHOE micromirrors. Although the authors analyze the recorded matrix of micromirrors in the form of Bragg holograms, the calculation is carried out in the language of geometric optics, which is strange. It is necessary to prove the applicability of this approach, for example, by comparing the obtained beam divergence after its interaction with the microhologram. Perhaps the authors believe that it is sufficient to present the experimental results in Fig. 5. But the illustration without linking its parameters to the parameters of the optical scheme is not very convincing.

In addition, there are purely editorial notes on the submitted manuscript:

  1. Formulas in the text are not numbered, which makes it difficult to refer to them, for example, when reviewing or when citing the text in the future.
  2. Line 223 (Figure ?? shows reconstructions for calibrations using both approaches) does not include a figure number.
  3. In the caption to Fig. 2 and fig. A1 there is no description of the vector n̄
  4. Also, in the caption to fig. A1 there is no description of points C and C '. These are probably the same points as in Fig. 2, but this should be said in the figure caption.
  5. It is not indicated in Fig. 2 and Fig. A1 that n Ì„ is the normal vector to the microlens surface. Therefore, the thought of the authors, when they unfold a spherical surface into a plane, is difficult to read. It seems that the legitimacy of this operation must be proven, and here you simply cannot do without complete designations.

In general, the manuscript is of interest to developers of 3D displays and can be published after minor revisions.
